# Allosteric activation of cell wall synthesis during bacterial growth

Irina Shlosman[1], Elayne M. Fivenson[2], Morgan S. A. Gilman[1], Tyler A. Sisley [2], Suzanne Walker [2], Thomas G. Bernhardt[2,3], Andrew C. Kruse [1] & Joseph J. Loparo [1]

The peptidoglycan (PG) cell wall protects bacteria against osmotic lysis and determines cell shape, making this structure a key antibiotic target. Peptidoglycan is a polymer of glycan chains connected by peptide crosslinks, and its synthesis requires precise spatiotemporal coordination between glycan polymerization and crosslinking. However, the molecular mechanism by which these reactions are initiated and coupled is unclear. Here we use single-molecule FRET and cryo-EM to show that an essential PG synthase (RodA-PBP2) responsible for bacterial elongation undergoes dynamic exchange between closed and open states. Structural opening couples the activation of polymerization and crosslinking and is essential in vivo. Given the high conservation of this family of synthases, the opening motion that we uncovered likely represents a conserved regulatory mechanism that controls the activation of PG synthesis during other cellular processes, including cell division.

Peptidoglycan (PG) is the major load-bearing structure of the bacterial cell wall, and its assembly and maintenance are critical for the shape determination and survival of virtually all bacteria[1]. As animal cells lack this structure entirely, PG synthesis is an attractive target for antibiotic development[2]. PG is produced from the precursor lipid II via two enzymatic steps: (1) glycosyltransferases (GT) polymerize lipid II into glycan chains; (2) transpeptidases (TP) crosslink the nascent chains to the existing PG mesh[3–5]. Polymerization and crosslinking reactions must be precisely coordinated to produce PG of the desired architecture and mechanical properties, as well as to avoid futile enzymatic cycles. Indeed, uncoupling between these reactions was shown to underlie the lethal action of penicillin-like antibiotics that induce futile cycles of glycan chain synthesis and degradation and tax bacteria energetically[6]. Despite the central importance of enzymatic coupling, the regulatory mechanisms that ensure spatiotemporal control of these reactions are largely unknown.

Most bacteria rely on two distinct systems to catalyze polymerization and crosslinking: the bifunctional class A penicillin-binding proteins (aPBPs) and the SEDS (shape, elongation, division, sporulation) glycosyltransferases that work in tandem with their cognate class

B PBP (bPBP) transpeptidases[7–11]. Class A PBPs are thought to be involved primarily in PG fortification and repair, while SEDS-bPBP synthases function in the context of two evolutionarily related multi-protein complexes to accomplish directed PG synthesis during cell elongation (the Rod complex/elongasome) and division (the divisome)[12–19]. The core of the divisome is comprised of the PG synthase FtsW-FtsI and accessory components FtsQLB and FtsN, while the Rod complex has a PG synthase RodA-PBP2 and accessory components MreCD and RodZ. RodA-PBP2 and FtsWI-FtsI are highly homologous and likely share key molecular details of function. Both FtsW and RodA require their cognate bPBP partners to polymerize in vitro, suggesting that polymerization and crosslinking reactions are allosterically coupled within these complexes[10,11]. Recent structural work on the model synthase RodA-PBP2 from *Thermus thermophilus* proposed that PBP2 might regulate RodA through periplasmic contacts that flank the active site of RodA (interface II)[11] (Fig. 1a, top). In broad agreement with the crystal structure, genetic studies reported a host of interface II variants that alter FtsW and RodA polymerization activities and implicated their respective accessory components, FtsQLB and MreC, as potential regulators of enzymatic activity[10,11,20–22]. However, given that both

[1]Department of Biological Chemistry and Molecular Pharmacology, Blavatnik Institute, Harvard Medical School, Boston, Massachusetts 02115, USA. [2]Department of Microbiology, Blavatnik Institute, Harvard Medical School, Boston, Massachusetts 02115, USA. [3]Howard Hughes Medical Institute, Harvard Medical School, Boston, Massachusetts 02115, USA. e-mail: andrew_kruse@hms.harvard.edu; joseph_loparo@hms.harvard.edu

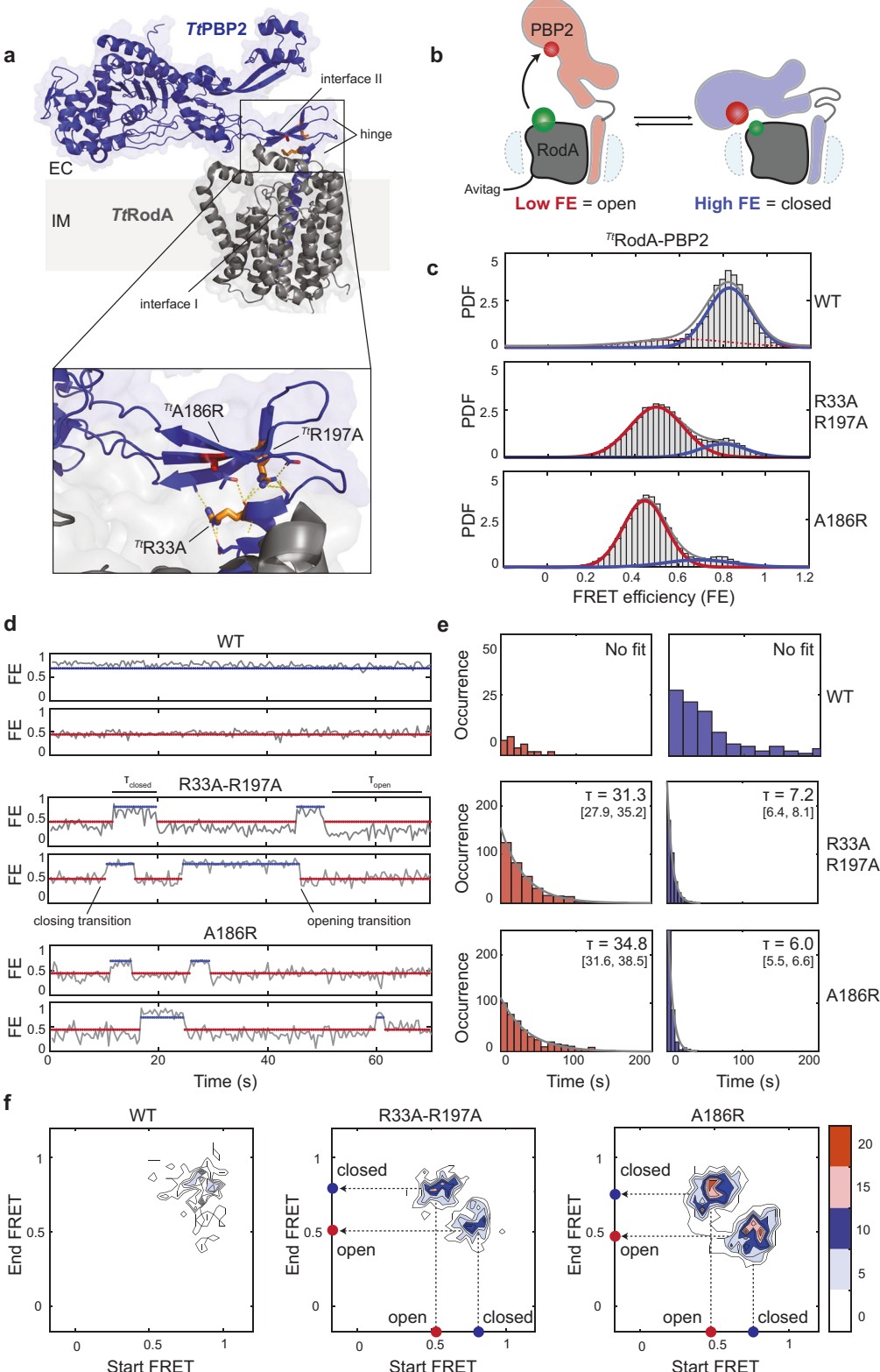

inhibitory and activating mutations were identified, in some cases at neighboring positions, the role of this interface in the regulation of polymerization remains unclear.

Paradoxically, the crystal structure also revealed that periplasmic contacts with RodA likely preclude PBP2 from accessing the PG matrix in cells. This conformation termed the closed state, positions the TP domain $^{Tt}$PBP2 within 40–50 Å of the inner membrane—out of reach of the PG layer, which lies 80–100 Å above the inner membrane in Gram-

negative bacteria[23–25]. By comparison, the TP site of an aPBP from *Escherichia coli* (PBP1b) is 70 Å above the inner membrane, suggesting that this degree of elevation is required to reach the PG[26,27]. The apparent geometric mismatch between class A and class B PBPs raises the question as to how PBP2 is able to carry out crosslinking while regulating RodA through interface II.

Here we show that enzymatic regulation of the RodA-PBP2 synthase is achieved through the structural dynamics of PBP2. Using

**Fig. 1 | smFRET analysis of $^{Tt}$RodA-PBP2$^{WT}$ and dynamic mutants. a** $^{Tt}$RodA-PBP2 X-ray structure (PDB: 6pl5) showing R33A, R197A (orange), and A186R (red) mutations as sticks; hydrogen bonds shown as dashed lines. The extracellular loop of RodA is omitted from the inset for clarity. **b** Schematic illustrating the smFRET assay, with one of the two possible orientations of donor (green) and acceptor (red) labels shown for simplicity. In the closed state, the labels are within 30–40 Å, which leads to high FRET efficiency (FE), while an increase in distance between labels upon structural opening leads to a decrease in FE. **c** Probability density (PDF) histograms of FE values derived from single-molecule trajectories for $^{Tt}$WT, $^{Tt}$R33A-R197A, and

$^{Tt}$A186R (pSI7, pSI12, pSI13). Normal fits to the data are shown in red (low-FRET), blue (high-FRET), and gray (compound), assuming a two-state model.
**d** Representative trajectories from (**c**), showing transitions between low-FRET (red) and high-FRET (blue) states. Markers are plotted at the mean values of the states as in Supplementary Table 1. **e** Dwell-time histograms and exponential fits for the low-FRET (red) and high-FRET (blue) states. Mean dwell times alongside 95% confidence intervals for all populations are indicated on the plots. **f** Transition density plots, normalized to the total observation time, show the frequency of closing and opening transitions for imaging constructs.

a combination of single-molecule FRET (smFRET) and cryo-EM, we demonstrate that *E. coli* RodA-PBP2 undergoes large-scale conformational rearrangements that enhance polymerization and facilitate crosslinking by elevating PBP2 above the inner membrane. Mutations that disrupt interface II contacts dramatically increase the frequency of structural opening and rescue shape defects in cells. Thus, interface II acts as an inhibitor of catalysis, repressing polymerization and sequestering PBP2 away from its PG substrate, while structural opening relieves this inhibition, activating both enzymes for concordant PG synthesis. This allosteric framework ensures that RodA and PBP2 are directly coupled to prevent futile enzymatic cycles and explains how initiation and activation of cell wall synthesis are achieved in vivo.

## Results

### Single-molecule imaging of $^{Tt}$RodA-PBP2

To measure the structural dynamics of the RodA-PBP2 complex, we developed a single-molecule FRET (smFRET) TIRF imaging assay. In brief, we introduced an AviTag at the N-terminus of $^{Tt}$RodA to conjugate the complex to the passivated surface of a microfluidic chamber and cysteine mutations ($^{Tt}$RodA$^{N195C}$, $^{Tt}$PBP2$^{K403C}$) for fluorophore labeling at positions that are sensitive to structural rearrangements of the TP domain (Fig. 1b, also "Methods"). In the closed conformation observed in the crystal structure, the two labels are within 30–40 Å of each other, corresponding to a high-FRET-efficiency state (high-FRET), while any opening motion was expected to increase the distance between them, reducing FRET efficiency (low-FRET) (Fig. 1b). The resulting construct ($^{Tt}$RodA-PBP2$^{WT}$) expressed and purified efficiently, labeled specifically with sulfo-Cy3 and sulfo-Cy5 fluorophores, and retained catalytic activity after labeling (Supplementary Fig. 1, also "Methods").

Single-molecule imaging of $^{Tt}$RodA-PBP2$^{WT}$ showed a major population centered at FRET efficiency ~0.8 (Fig. 1c, blue curve), consistent with the closed conformation of $^{Tt}$RodA-PBP2 in the X-ray structure ("Methods"). A long tail of sparsely populated lower-FRET states hinted at the presence of alternative, less energetically accessible conformations (Fig. 1c, red dashed curve); however, the fraction of trajectories that visited a low-FRET state constituted less than 2% of the dataset, such that this population could not be fitted with confidence (Fig. 1c, d, also "Methods"). Dwell-time histograms showed that the high-FRET state was stable on the timescales of tens to hundreds of seconds, while excursions into the low-FRET state were rare and short-lived (Fig. 1d, e, also "Methods"). Finally, transition plot analysis, which depicts the frequency of transitions between the two states as a heatmap, revealed that few real transition events were detected for $^{Tt}$RodA-PBP2$^{WT}$ (Fig. 1f, Supplementary Table 1). These results indicate that apo-$^{Tt}$RodA-PBP2 exists predominantly in the closed state and is largely static, in contrast to previous exploratory negative stain imaging that suggested substantial conformational heterogeneity of this complex[11].

### Hinge and interface mutations promote structural dynamics in $^{Tt}$PBP2

To determine what contacts within $^{Tt}$RodA-PBP2$^{WT}$ contribute to the structural stability of the closed state, we introduced mutations in two

regions of the complex: (1) at interface II ($^{Tt}$PBP2$^{A186R}$) and (2) within the hinge region ($^{Tt}$PBP2$^{R33A-R197A}$), where a tight hydrogen-bond network prevents TP domain opening (Fig. 1a, inset, Supplementary Fig. 2). Interface and hinge mutants were found to be fully functional and exhibited markedly shifted FRET ensembles compared to $^{Tt}$RodA-PBP2$^{WT}$, with a major stable low-FRET state, in addition to a minor and more transient high-FRET state (Fig. 1c–e, Supplementary Fig. 1). Moreover, mutant trajectories displayed frequent transitions between low- and high-FRET states, indicating that they were structurally dynamic (Fig. 1d–f, Supplementary Table 1). Our smFRET results demonstrate that $^{Tt}$RodA-PBP2$^{WT}$ exists primarily in the closed state but that mutations in the hinge and interface II regions promote large-scale structural rearrangements of the TP domain.

### Cryo-EM analysis of the A186R mutant reveals an open state in PBP2

To resolve the conformations visited by $^{Tt}$RodA-PBP2, we took advantage of cryo-electron microscopy (cryo-EM), a technique that can capture structural heterogeneity in near-native conditions. We analyzed two samples that are at the opposite extremes of the FRET spectrum: $^{Tt}$RodA-PBP2$^{WT}$ (high-FRET) and the interface mutant $^{Tt}$RodA-PBP2$^{A186R}$ (low-FRET). Cryo-EM 2D class averages of the $^{Tt}$WT complex showed a well-resolved TP domain with no visible conformational heterogeneity, whereas this region appeared flexible in the $^{Tt}$A186R mutant (Fig. 2a). Accordingly, ab initio reconstruction of $^{Tt}$WT particles yielded a 3D map that recapitulated the X-ray structure (Fig. 2b, Supplementary Fig. 3). The $^{Tt}$A186R particles, on the other hand, were classified into two distinct 3D maps: one similar to the closed state of $^{Tt}$WT, and an alternative state in which the TP domain of $^{Tt}$PBP2 is raised straight up (Fig. 2c, Supplementary Fig. 3, also "Methods"). We estimate from the size of the TP domain that the catalytic site of $^{Tt}$PBP2 in the open state lies roughly 70–80 Å above the inner membrane—sufficiently elevated to reach the PG layer in vivo. The open and closed maps are consistent with the distances expected for the low- and high-FRET states, respectively, and confirm that transitions between these states correspond to TP domain motion ("Methods"). Due to the considerable conformational dynamics of the $^{Tt}$A186R mutant, the structure of the open state could not be resolved at the atomic level (Supplementary Table 2). However, the overall orientation of the protein in this conformation was consistent with the AlphaFold model of $^{Tt}$RodA-PBP2, allowing us to explore the mechanism driving the opening motion[28,29] (Fig. 2d, Supplementary Fig. 4). Comparing the AlphaFold prediction to the crystal structure, we found that upon opening, the hinge region of $^{Tt}$PBP2 is predicted to undergo a secondary structure transition from a coil to a short α-helix, which effectively breaks interface II contacts and elevates the TP domain above the inner membrane (Fig. 2d, Supplementary Fig. 4). This potential mechanism is in line with our observation that structural dynamics of the TP domain are accelerated by mutations in the hinge and interface regions.

### *Escherichia coli* RodA-PBP2 is structurally dynamic

WT RodA-PBP2 from *T. thermophilus* was found to be static, likely due to the fact that for a thermophile, the energetic barrier to the opening transition is too high at 25 °C. Therefore, we hypothesized

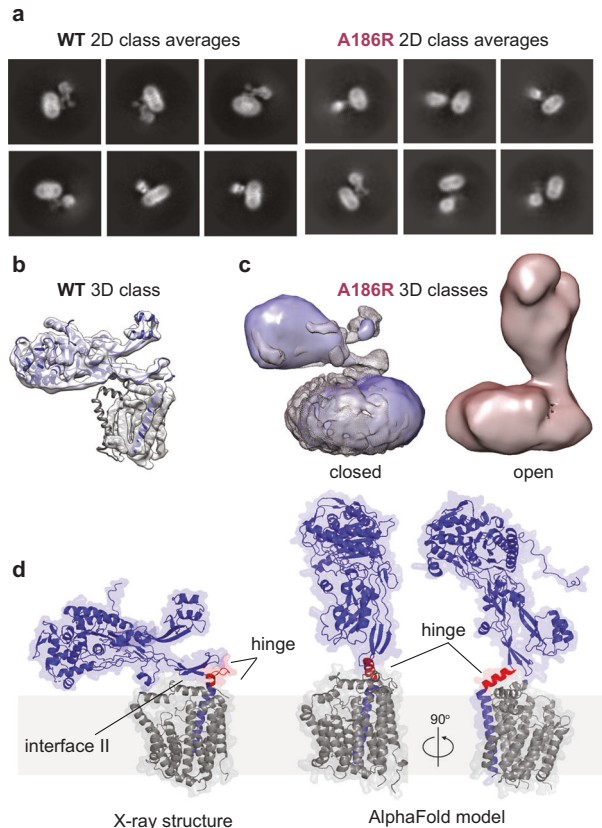

**Fig. 2 | Cryo-EM analysis reveals an open state in $^{Tt}$RodA-PBP2$^{A186R}$.**
**a** Representative 2D class averages for $^{Tt}$RodA-PBP2$^{WT}$ (pMS239) and the $^{Tt}$RodA-PBP2$^{A186R}$ mutant (pMS293). Each class contains 1–2k ($^{Tt}$WT) and 2–3k ($^{Tt}$A186R) particles. **b** $^{Tt}$WT X-ray structure (PDB: 6pl5) modeled into 3D reconstruction of $^{Tt}$WT particles (gray volume). **c** Left: Comparison of $^{Tt}$WT map (gray) with 3D reconstruction of the closed state of the $^{Tt}$A186R mutant (blue). Right: open state of $^{Tt}$A186R (red). Closed and open maps were resolved with 3D variability analysis. **d** Comparison of X-ray structure and AlphaFold model of $^{Tt}$RodA-PBP2$^{WT}$. The hinge region (red) is a coil in the structure and helical in the model.

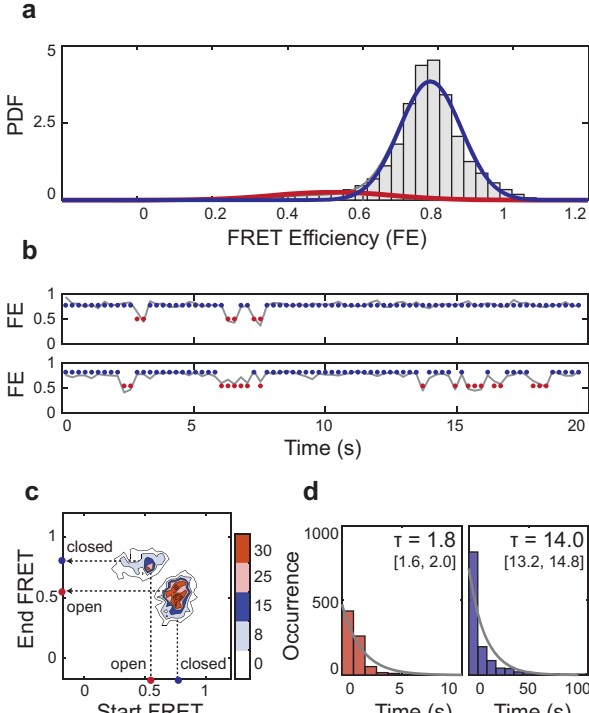

**Fig. 3 | $^{Ec}$RodA-PBP2$^{Mon}$ undergoes structural opening in vitro. a** Probability density (PDF) histogram of FE values derived from single-molecule trajectories for $^{Ec}$RodA-PBP2$^{Mon}$ (pSI126). Normal fits to the data are shown in red (low-FRET), blue (high-FRET), and gray (compound), assuming a two-state model. **b** Two representative trajectories from (**a**), showing transitions between high- and low-FRET states. Markers are plotted at the mean values of the two states, as in Supplementary Table 1. **c** Transition plot analysis shows the frequency of closing and opening transitions of $^{Ec}$Mon. **d** Dwell-time histograms of the low- and high-FRET states show that the low-FRET state in *E. coli* WT is a detectable yet transient conformation.

that a mesophilic homolog of RodA-PBP2 from *E. coli*, whose kinetic regime is finely tuned to moderate temperatures, might undergo stochastic opening under laboratory conditions. To produce *E. coli* imaging constructs, we introduced labels at homologous positions to those of *T. thermophilus* ($^{Ec}$RodA$^{L201C}$ and $^{Ec}$PBP2$^{R425C}$). Given the high degree of conservation between the two homologs, this strategy allowed us to compare their smFRET populations directly. Because $^{Ec}$RodA-PBP2 formed non-physiological dimers when overexpressed in vitro, we introduced two monomerizing mutations in $^{Ec}$PBP2 ($^{Ec}$PBP2$^{V143T}$ and $^{Ec}$PBP2$^{A147S}$) at positions far removed from the active site to improve the biochemical behavior of the sample (Supplementary Fig. 5A–C, also "Methods"). The resulting construct ($^{Ec}$RodA-PBP2$^{Mon}$/ $^{Ec}$Mon) had robust catalytic activity and superior biochemical properties, allowing us to carry out smFRET imaging experiments (Supplementary Fig. 5D, E). These experiments revealed that the conformational ensemble of $^{Ec}$Mon was qualitatively similar to that of $^{Tt}$WT, with a major high-FRET state and a minor low-FRET state (Fig. 3a). In contrast to *T. thermophilus*, however, the *E. coli* homolog was dynamic, transitioning rapidly between the two states (Fig. 3b, c). While the low-FRET state was considerably shorter-lived than the high-FRET state, it was readily detectable by our measurements (Fig. 3d), demonstrating that $^{Ec}$Mon spontaneously adopted a transient open state. This result supports our hypothesis that PBP2 undergoes conformational exchange that brings its active site close to the PG mesh, promoting crosslinking.

## Structural opening of PBP2 promotes RodA polymerization activity

If the open state of PBP2 represents the crosslinking-competent state, then what is the effect of structural opening on RodA polymerization activity? We had previously suggested that modifications at interface II in *T. thermophilus* RodA-PBP2 ($^{Tt}$RodA-PBP2$^{A186R}$, $^{Tt}$RodA-PBP2$^{L43R}$) might have an antagonistic effect on polymerization, whereas an interface II mutant in *E. coli* ($^{Ec}$RodA-PBP2$^{L61R}$) was shown to be hyperactive[11,22]. To clarify the effect of these changes definitively, we carefully probed the polymerization activity of the two dynamic mutants from *T. thermophilus* ($^{Tt}$R33A-R197A, $^{Tt}$A186R) along with $^{Tt}$L43R. In the process of optimizing the conditions of the polymerization assay, we discovered that small variations in detergent levels, which frequently arise when concentrating a protein sample in detergent, dramatically impacted polymerization activity (Supplementary Fig. 6A). When we assayed the proteins via an optimized protocol that controls for detergent concentration, we found that both dynamic mutants had enhanced polymerization activity: they produced longer glycan chains, as well as more overall product, than $^{Tt}$WT (Fig. 4a, also "Methods"). $^{Tt}$L43R had reduced activity, as previously reported, and here we observed that this mutation led to the dissociation of the complex upon purification by size-exclusion chromatography, which would account for its polymerization defect (Supplementary Fig. 6B). Thus, mutations that disrupt interface II enhance RodA polymerization activity.

To confirm that this allosteric mechanism is general and not specific to the selected mutations, we devised an alternative strategy

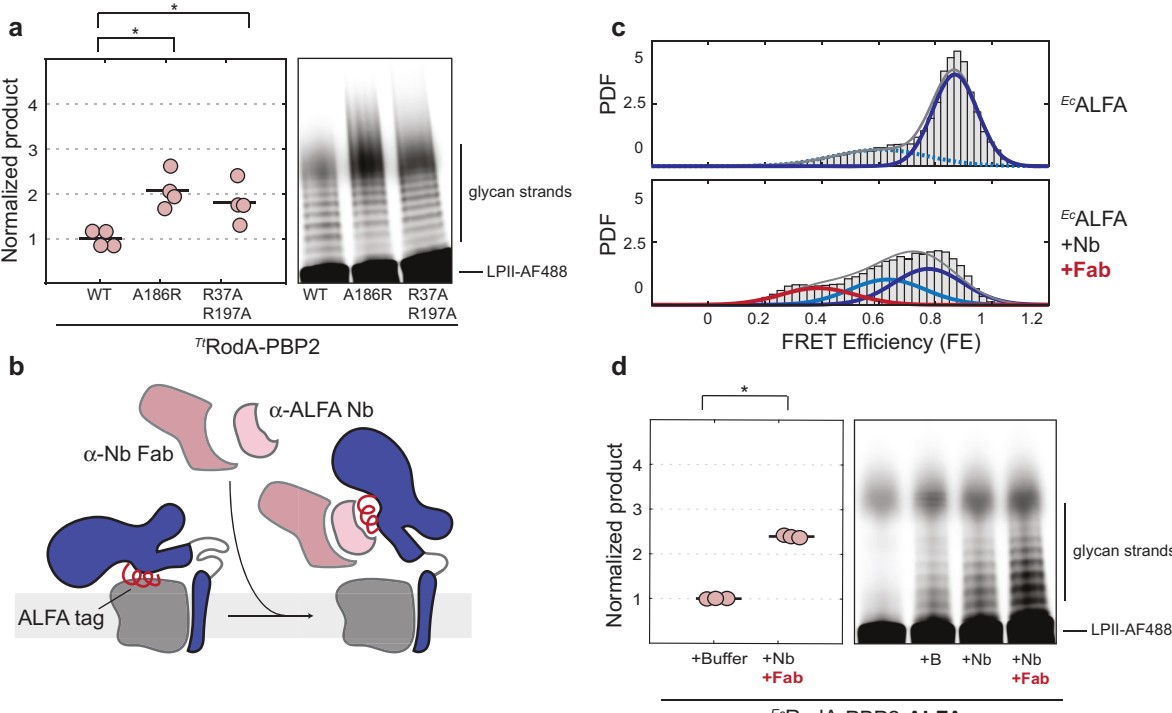

**Fig. 4 | Structural opening promotes RodA polymerization activity. a** Dynamic mutants have enhanced polymerization activity. Polymerization assay (right) and quantification plot (left) of $^{Tt}$RodA-PBP2$^{WT}$, $^{Tt}$A186R, and $^{Tt}$R33A-R197A (pSI7, pSI12, pSI13). Individual technical replicates ($n = 4$) are shown as pink circles, and the averages are shown as black lines. The image shown is representative of $n = 2$ independent experiments. Pairwise $p$-values were calculated using a two-sample two-sided $t$-test as described in "Methods": 0.007 ($^{Tt}$WT vs. $^{Tt}$A186R), 0.032 ($^{Tt}$WT vs. $^{Tt}$R33A-R197A). **b** Schematic illustrating how the binding of anti-ALFA nanobody and Fab to the ALFA-tag at interface II stabilizes the open state of $^{Ec}$RodA-PBP2$^{ALFA}$. **c** Population density (PDF) histograms of FE values derived from single-molecule

trajectories of $^{Ec}$ALFA (pSI147) imaged either with or without 2 μM Nb-Fab complex. Normal fits to the data are shown in red (low-FRET), light blue (middle-FRET), blue (high-FRET), and gray (compound), assuming a two-state model for $^{Ec}$ALFA and a three-state model for Fab-bound $^{Ec}$ALFA. **d** Polymerization assay (right) and quantification plot (left) of $^{Ec}$ALFA with buffer control (+B), 2 μM Nb or 2 μM Nb-Fab complex. Individual technical replicates ($n = 3$) are shown as pink circles, and the averages are shown as black lines. The image shown is representative of $n = 2$ independent experiments. Analysis of the polymerization assay was carried out as in (**a**) to obtain a $p$-value of 6.3e-5 (buffer vs. Nb-Fab condition).

to stabilize the open state of PBP2 via a conformationally selective nanobody. Specifically, we swapped a loop at interface II of $^{Ec}$RodA-PBP2, which is dispensable for complex stability and activity (Supplementary Fig. 6C), with a 15-amino-acid ALFA-tag sequence ($^{Ec}$RodA-PBP2$^{ALFA}$/ $^{Ec}$ALFA). This tag is recognized by the anti-ALFA nanobody (Nb), which in turn can be coupled to a nanobody-binding Fab[30,31] (Fig. 4b, also "Methods"). In this setup, the binding of the Nb-Fab complex to $^{Ec}$ALFA was expected to favor the open state since, in the closed conformation, the recognition sequence is buried by interface II. Indeed, smFRET imaging of $^{Ec}$ALFA showed a single major high-FRET state, consistent with the idea that the engineered loop did not in and of itself disrupt the interface (Fig. 4c, Supplementary Fig. 6D–F). Upon addition of the Nb-Fab complex, the conformational equilibrium became heavily biased toward lower-FRET states, indicating that Nb-Fab binding favored structural opening (Fig. 4c, Supplementary Fig. 6D–F). We note that the Fab-containing sample exhibited two distinct lower-FRET populations, suggesting the presence of a partially open intermediate as well as the open state. Similar to the effect of interface II mutations, binding of the Nb-Fab complex dramatically increased the rate of polymerization relative to the control (Fig. 4d). Collectively, these results establish that structural opening of RodA-PBP2 activates polymerization.

### PBP2 dynamics are required for Rod complex function in vivo

Given that the open state is activated for polymerization and likely facilitates crosslinking, we hypothesized that structural dynamics are required for PG synthesis in vivo. A prediction of this hypothesis is that

disruption of the opening motion should lead to impaired PG synthesis, defects in cell viability, and aberrant morphology. To test this prediction, we designed a locked mutant of $^{Ec}$RodA-PBP2, in which the protein was constrained to the closed state with a disulfide bridge between two cysteines in the hinge region ($^{Ec}$PBP2$^{D49C-K240C}$) (Fig. 5a). The resulting construct retained expression, fold and substrate binding to the TP domain (Supplementary Fig. 7), and SDS-PAGE analysis confirmed the formation of a disulfide with 50% efficiency (Supplementary Fig. 7, also "Methods"). To test the effect of the conformational lock on Rod complex activity, we carried out viability experiments in a *rodA-pbpA* deletion strain, complementing the deletion with WT $^{Ec}$RodA and $^{Ec}$PBP2 variants on IPTG-inducible plasmids (Fig. 5b). Cells expressing the locked mutant showed a reduction in viability compared to WT and single-cysteine controls (Fig. 5c, LB +IPTG), as well as shape defects, consistent with the inefficient rescue of the *rodA-pbpA* deletion (Fig. 5d). We attribute the residual activity observed in the $^{Ec}$RodA-PBP2$^{D49C-K420C}$ strain to the population of unlocked PG synthases (~50%). Notably, viability and shape defects were fully rescued upon the addition of a reducing agent (Fig. 5c, d, +DTT), confirming that the conformational lock, not the cysteine modifications themselves, impaired Rod complex activity. From this, we conclude that the open state of $^{Ec}$RodA-PBP2 is required for PG synthesis in vivo.

### MreC likely regulates TP domain dynamics in vivo

Despite its key role in vivo, the open state of $^{Ec}$RodA-PBP2 is a minor and transient conformation in vitro (Fig. 3). This suggests that in a

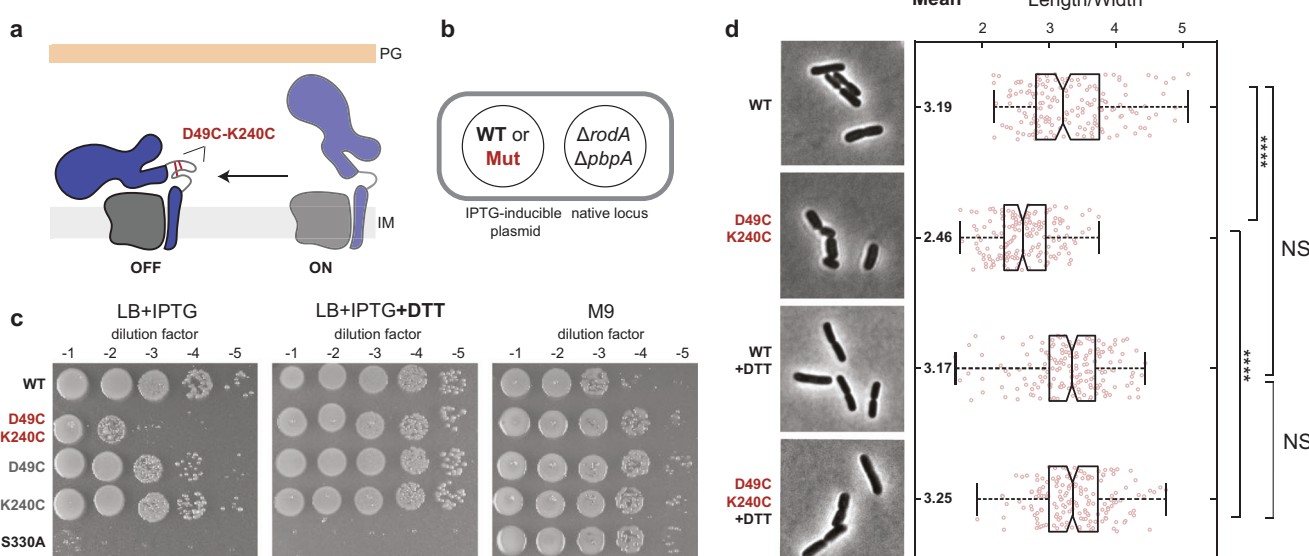

**Fig. 5 | *Ec*RodA-PBP2 structural dynamics are critical for Rod complex function in vivo. a** Schematic showing the proposed inhibitory effect of the disulfide lock (D49C-K240C) on the structural dynamics and function of *Ec*RodA-PBP2. **b** Schematic of the complementation assay. **c** Conformational lock severely reduces complementation efficiency. Deletion strains bearing WT *Ec*RodA-PBP or PBP2 variants were spotted either on LB plates (Rod complex essential) with or without DTT or on M9 plates (Rod complex nonessential). *Ec*RodA-PBP2^S330A (TP-deficient mutant) was used as a negative control. The titers shown are representative of three biological replicates. **d** Conformational lock impairs shape. Representative micrographs of *Ec*WT and *Ec*D49C-K240C strains from (**c**), grown either with or without 10 mM DTT. Length-to-width ratio is plotted for cells in each condition alongside summary statistics. The central mark on the boxplot indicates the median value, while the edges of the box mark the 25% and 75% percentiles. The whiskers show the most extreme values of each dataset. Two-sample, two-sided *t*-test was used to determine pairwise *p*-values between conditions: 4.37e-28 (*Ec*WT vs. *Ec*D49C-K240C), 0.19 (*Ec*WT with DTT vs. *Ec*D49C-K240C with DTT), 0.84 (*Ec*WT vs. *Ec*WT with DTT), 4.58e-39 (*Ec*D49C-K240C vs. *Ec*D49C-K240C with DTT). The data are representative of two biological replicates.

physiological context, *Ec*RodA-PBP2 dynamics might be regulated to promote efficient initiation of PG synthesis, either by a substrate or an accessory factor. A likely candidate for this role is MreC, an essential component of the Rod complex that was shown to bind PBP2 through contacts in the pedestal domain[32] (Fig. 6a, left). Suggestively, single amino-acid substitutions in *Ec*RodA-PBP2 (*Ec*PBP2^L61R, *Ec*PBP2^T52A) that bypass the requirement for functional MreC in vivo[22] map to the hinge and interface II regions that we identified as key energetic barriers to the opening transition in *T. thermophilus* (Fig. 6a, inset, Supplementary Fig. 2). Finally, AlphaFold modeling of the ternary complex between *Ec*RodA, *Ec*PBP2, and *Ec*MreC predicts that MreC binds the open conformation of PBP2 (Supplementary Fig. 8). Together, these data point to a mechanism in which MreC binds to the pedestal domain of PBP2, enhancing structural opening and activating catalysis.

Because we were unable to purify a monodisperse preparation of *Ec*MreC suitable for use in smFRET assays, we instead investigated the effect of PBP2 bypass alleles on structural dynamics as a proxy for MreC activity (Fig. 6b, Supplementary Fig. 9A). In line with our predictions, both *Ec*RodA-PBP2^T52A and *Ec*RodA-PBP2^L61R exhibited a shift to the open state (Fig. 6b) and had enhanced polymerization activity compared to *Ec*RodA-PBP2^Mon (Fig. 6c). Rather than stabilize the open state, these mutants appeared to lower the energetic barrier to the opening transition, inducing rapid exchange between the two states (Supplementary Fig. 9B, C, also "Methods"). Strikingly, the degree to which each suppressor mutant enhanced structural opening and polymerization activity in vitro paralleled the ability of that mutant to rescue defects of a dysfunctional MreC variant (*Ec*MreC^R292H) in vivo[22]. *Ec*RodA-PBP2^L61R, which fully restored shape in cells producing *Ec*MreC^R292H, was hyperactive in vitro relative to the partial suppressor *Ec*T52A, further reinforcing the notion that structural dynamics are correlated with activity.

If structural opening is necessary and sufficient to account for the suppressor phenotype of *Ec*L61R and *Ec*T52A mutants, then any mutant that stabilizes the open state of *Ec*RodA-PBP2 should recapitulate their effect—irrespective of the molecular mechanism by which conformational bias is induced. To test this prediction, we designed an open mutant using an orthogonal strategy: rather than disrupting the closed state, as was done previously, we engineered mutations that would specifically favor the open state. Guided by the AlphaFold model of *Ec*PBP2, we introduced two mutations in the hinge region that stabilize the alpha-helical conformation predicted in the open state with salt bridges (*Ec*T52R bonded to *Ec*E56 and *Ec*N55E bonded to *Ec*R58) (Fig. 6a, right). A third mutation (*Ec*S54A) removed a hydrogen bond with *Ec*N55 to further favor the formation of the salt bridge between *Ec*N55E and *Ec*R58. The triple mutant (*Ec*RodA-PBP2^T52R·S54A·N55E) was found to be fully in the open state, with no detectable closed population (Fig. 6b, Supplementary Fig. 9B–D) and exhibited even more robust polymerization activity than either *Ec*T52A or *Ec*L61R (Fig. 6c). Notably, the efficiency with which the triple mutant rescued cell viability and shape defects in *Ec*MreC^R292H background was dramatically increased relative to *Ec*T52A (Fig. 6d, f), supporting the model in which MreC regulates structural dynamics in vivo. The triple mutant also sensitized *E. coli* cells to sub-MIC concentrations of mecillinam much more strongly than *Ec*T52A, indicating that this mutant had elevated Rod complex activity (Fig. 6e). Collectively, these results demonstrate that the open conformation that we uncovered represents the active state of *Ec*RodA-PBP2 and that bacterial viability and shape can be precisely tuned by altering the conformational ensemble of this enzyme.

## Discussion

Precise spatiotemporal coordination between polymerization and crosslinking is required to produce PG of the correct architecture, as well as to prevent futile cycles of glycan synthesis and degradation that are toxic to bacteria[6,33]. Our findings reveal that enzymatic activation and coupling are achieved through conformational rearrangements in RodA-PBP2. Structural opening of the TP domain acts as an allosteric switch, simultaneously activating polymerization and elevating the active site of PBP2 to enable crosslinking, which ensures that

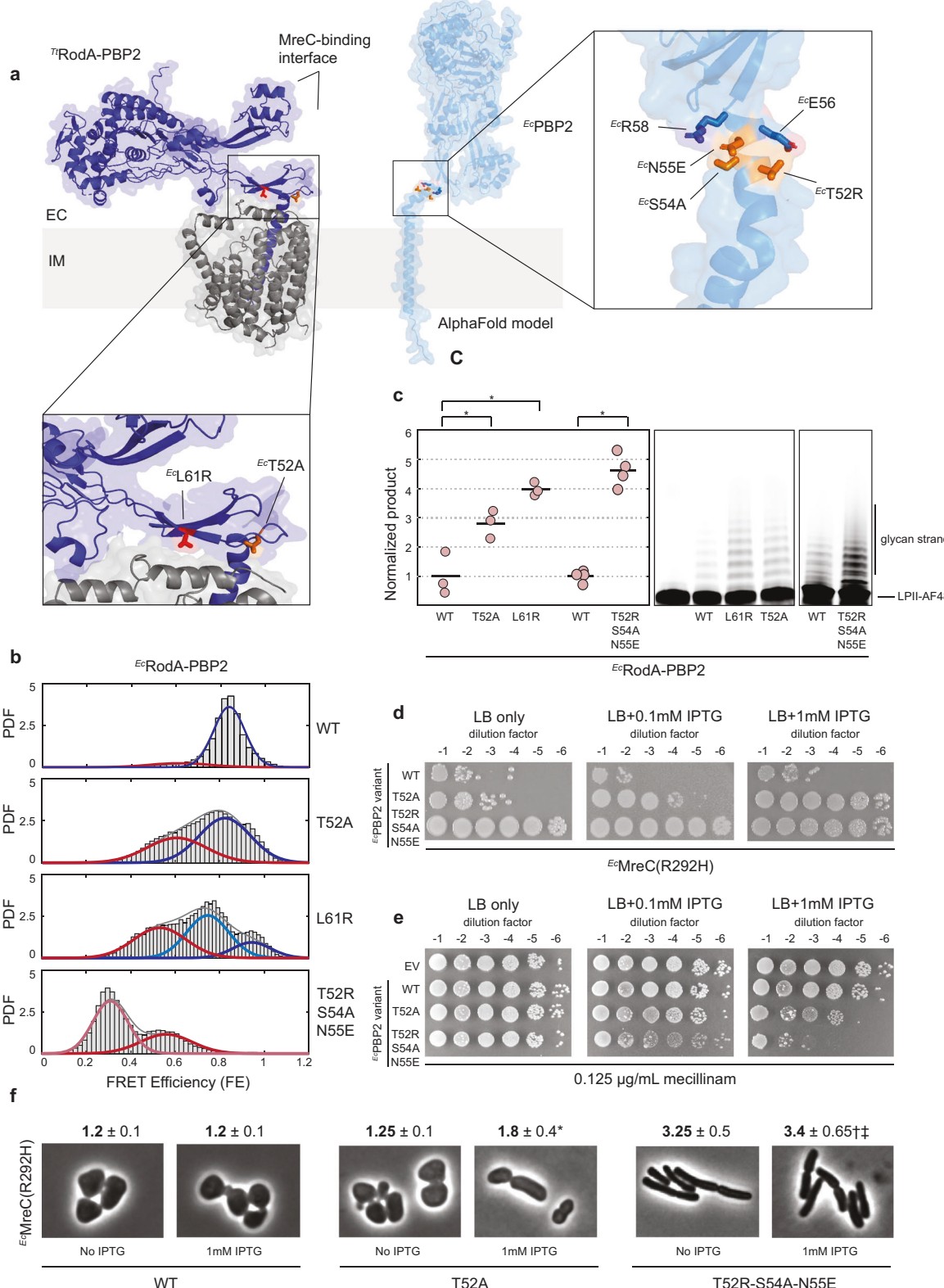

**f** <sup></sup>

unproductive catalysis is minimized. Altering the conformational ensemble of RodA-PBP2 to the open or the closed state dramatically affects cell viability and morphology–either promoting or inhibiting growth–highlighting the central role that this conformational switch plays in the initiation and regulation of PG synthesis in vivo.

In our allosteric model, PBP2 regulates RodA through conformational motion rather than through direct coupling between the two

enzymatic sites (Fig. 7). We do not exclude the possibility of enzymatic crosstalk between GT and TP sites, e.g., mediated by the nascent glycan chain. However, given that crosslinking is inhibited in our activity assays, structural rearrangements alone can account for changes in polymerization activity that we observe. The two states of PBP2 allow it to act both as an activator of polymerization by RodA through interface I and as an inhibitor through interface II (Fig. 7). Structural

**Fig. 6 | Mutants that induce structural opening suppress viability and shape defects of $^{Ec}$MreC$^{R292H}$. a** Left: $^{Ec}$MreC$^{R292H}$ suppressor mutations ($^{Ec}$PBP2$^{L61R}$ and $^{Ec}$PBP2$^{T52A}$) mapped onto the X-ray structure of $^{Tt}$RodA-PBP2 (PDB: 6pl5) at homologous positions (interface in red; hinge in orange). $^{Ec}$MreC-binding interface is predicted from the $^{Hp}$MreC-PBP2 X-ray structure (PDB: 5lp5). Right: AlphaFold prediction of $^{Ec}$RodA-PBP2$^{WT}$, showing mutations that stabilize alpha-helical hinge region as orange sticks and residues forming salt bridges as blue sticks. **b** Probability density (PDF) histograms of FE values derived from single-molecule trajectories for $^{Ec}$RodA-PBP2$^{WT}$ (reproduced from Fig. 3a) and dynamic mutants (pSI126, pSI127, pSI149, pSI152). Normal fits to the data are shown in pink (lowest-FRET), red (low-FRET), light blue (middle-FRET), blue (high-FRET), and gray (compound), assuming either a two- or three-state model. **c** Polymerization assays (right) and quantification (left) of suppressor mutant activity (same constructs as B). Technical replicates are in pink; averages are shown as black lines. These data are representative of $n = 2$ independent experiments. Analysis carried out as outlined in "Methods" to obtain $p$-values: 0.012 ($^{Ec}$WT-$^{Ec}$L61R), 0.027 ($^{Ec}$WT-$^{Ec}$T52A), and

0.004 ($^{Ec}$WT-$^{Ec}$T52R-S54A-N55E). **d** Strains bearing the $^{Ec}$MreC$^{R292H}$ mutation and different $^{Ec}$RodA-PBP2 variants were grown in M9 media (permissive) and spotted on LB plates (restrictive) with varying concentrations of IPTG. The images shown are representative of $n = 2$ independent experiments. **e** Mecillinam sensitivity titers of different $^{Ec}$RodA-PBP2 variants in the WT background show that the triple mutant sensitizes *E. coli* to mecillinam. Strains were grown ON in LB (permissive) and spotted on LB plates supplemented with sub-MIC concentration of mecillinam (restrictive if Rod complex activity is hyperactive). The images shown are representative of $n = 2$ independent experiments. **f** Morphology experiments with strains from (**d**), showing representative micrographs and quantification of length-to-width ratio for induced and uninduced conditions. At least 50 cells were analyzed for each condition, and $p$-values between relevant conditions were determined: *1.1e-14 for $^{Ec}$WT with IPTG vs. $^{Ec}$T52A with IPTG; †2.6e-37 for $^{Ec}$WT with IPTG vs. $^{Ec}$T52R-S54A-N55E with IPTG; ‡1.9e-32 for $^{Ec}$T52A with IPTG vs. $^{Ec}$T52R-S54A-N55E with IPTG.

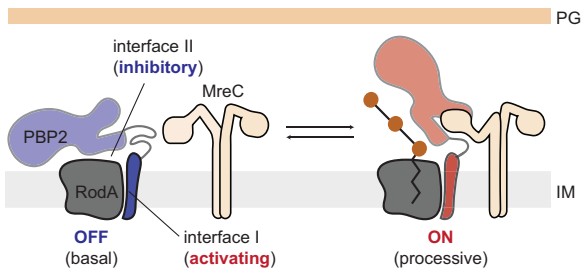

**Fig. 7 | Structural dynamics of PBP2 regulate PG synthesis.** Schematic view of the proposed model of PG synthesis, in which MreC controls enzymatic activation through structural dynamics of PBP2.

dynamics toggle PBP2 between these two modalities by disrupting and reforming contacts at interface II. While there are no substrate-bound structures of SEDS-bPBPs to explain how the two interfaces regulate polymerization on the molecular level, we propose that interface I helps form a lipid II binding site, while contacts at interface II prevent catalysis. Interface II could inhibit polymerization either indirectly, e.g., by sequestering critical contacts that become available in the open state, or directly, e.g., by sterically blocking one of the reaction steps. Recent structural work on the distantly related O-antigen ligase WaaL supports the latter model[34]. The structure suggests that in this family of enzymes, lipid headgroups are brought together and ligated above the extracytoplasmic loops[34]. If a similar mechanism is at play in RodA, interface II might prevent polymerization by obstructing contacts between lipid II and the growing glycan chain.

In cells, enzymatic inhibition by interface II is likely to be relieved by MreC when the PG synthase is incorporated into the Rod complex (Fig. 7). Transcriptional control maintains expression levels of RodA and PBP2 at roughly 100–200 per cell in *E. coli*[35]. However, the number of actively synthesizing Rod complexes constitutes only a fraction of the total protein pool, raising the question as to whether free RodA-PBP2 molecules contribute to PG synthesis[14]. We show that uncomplexed synthases are predominantly in the autoinhibited closed state, with enzymatic activity reduced to basal levels, which would effectively safeguard the cell against undesirable catalysis outside of the Rod complex. While MreC is well conserved across bacterial taxa, some bacteria—notably polar growing actinomycetes like *Corynebacterium glutamicum* or *Mycobacterium tuberculosis*—lack the *mreCD* genes entirely[21,36,37]. Yet, the RodA-PBP2 synthase plays a key role in growth, suggesting a different means of activation[38–40]. One possibility is that some other accessory factors take on the role of MreC. Alternatively, interface II contacts may be weaker in those organisms, leading to higher structural dynamics. More generally, the finding that single amino-acid substitutions at interface II dramatically alter the conformational ensemble of a SEDS-bPBP—and by extension, the enzyme's activity—

suggests that this allosteric switch represents a powerful evolutionary adapter that can be tuned or repurposed to satisfy diverse molecular needs. This mechanism would account for how organisms that lack MreC initiate PG synthesis, as well as explain how some bacteria, like *Bacillus subtilis*, can achieve higher basal levels of SEDS activity[9].

SEDS proteins and their bPBP partners are nearly universally conserved, suggesting that the general features of allosteric coupling between GT and TP activities likely translate not only to RodA-PBP2 homologs in other organisms but also to the evolutionarily related PG synthase in the divisome[9]. The essential regulatory components of the divisome, FtsQLB, do not share sequence similarity with MreCD. However, functionally the FtsLB complex mimics the proposed role of MreC, i.e., hyperactivates polymerization by FtsWI, and the coiled-coil topology of the FtsLB heteromer is reminiscent of the MreC dimer[31,41,44–43]. Mutations that activate divisome activity in vivo map to interface II, pointing to a model in which FtsLB-induced structural dynamics of FtsI regulate PG synthesis in the divisome[19,41,44–47]. Finally, recent structural work on the divisome complex from *Pseudomonas aeruginosa* shows that FtsI adopts a partially open conformation when in complex with FtsQLB, further reinforcing the notion that the general features of this mechanism are conserved in the divisome[48]. The specific details of regulation, as well as the role of structural dynamics in cell division, remain important questions for future biophysical investigation.

Allosteric coupling between polymerization and crosslinking is a recurring motif in PG synthesis and is not unique to SEDS-bPBPs. Class A PBPs were shown to adopt distinct structural states, in which GT and TP domains are reoriented relative to each other, and mutations at the interface between the two domains render these enzymes insensitive to their physiological activators LpoA/B[26,49–52]. Thus, two orthogonal families of synthases—aPBPs and SEDS-bPBPs—appear to share a unifying regulatory framework in which a single conformational switch controlled by an accessory factor allows for precise activation of enzymatic activity. This biological logic ensures that futile cycles of PG synthesis are avoided and that synthesis by aPBPs and SEDS-bPBPs is spatially segregated into distinct physiological processes—PG fortification and repair versus cell division and elongation.

This work uncovered structural dynamics that govern the activity of RodA-PBP2, inspiring similar inquiry in other synthases and adding another dimension to the therapeutic space available to target cell wall biogenesis. Inhibitors that mimic or disrupt protein–protein interactions present a promising path for the development of conformationally selective drugs that enzymatically silence or hyperactivate PG synthesis.

## Methods
### Strains and plasmids
All in vivo strains used in this study (Supplementary Table 3) were derived from MG1655, FB38, and PR5[21,53,54]. Unless otherwise specified,

cells were grown either in LB medium at 37 °C or in M9 minimal medium supplemented with 0.2% maltose or glucose (abbreviated M9-malt or M9-glu) at 30 °C. Antibiotics were added as appropriate at 1:1000 dilution of the stocks: chloramphenicol (25 mg/mL), kanamycin (50 mg/mL), and tetracycline (10 mg/mL). RodA and PBP2 variants were produced via Gibson assembly (Supplementary Tables 4–6). All constructs were sequenced to confirm the incorporation of desired fragments or mutations. Strains with chromosomal deletions or mutations were constructed via transduction with bacteriophage P1 followed by selection on kanamycin.

### Titer experiments

For complementation experiments, *roda-pbpA* deletion strains (FB38) bearing either $^{Ec}$RodA-PBP2$^{WT}$ or PBP2 variants on IPTG-inducible plasmids (pHC857, pHC857, pSI47, pSI48, pSI49) were grown overnight in M9-malt-CAM (permissive) at 30 °C to $OD_{600} = 0.4–0.8$ AU. The cultures were spun down and resuspended in M9 to normalize $OD_{600}$ to 1 AU for all conditions. The resulting cultures were then serially diluted and spotted on either M9-malt-CAM agar plates (Rod complex nonessential) or LB-CAM agar plates (Rod complex essential) with 25 μM IPTG in the presence or absence of 10 mM DTT and incubated overnight at 37 °C. To ensure optimal rescue, DTT stocks and DTT-containing LB-agar plates were prepared fresh on the day of the experiment.

For $^{Ec}$MreC$^{R292H}$ suppressor experiments, strains bearing this mutation and supplemented with $^{Ec}$RodA-PBP2$^{WT}$ or $^{Ec}$RodA-PBP2 variants on IPTG-inducible plasmids (PR5/pHC857, PR5/pSI154, PR5/pSI157) were grown in M9-glu media supplemented with 0.2% casamino acids, serially diluted and spotted on LB-CAM plates with varying concentrations of IPTG for overnight incubation at 37 °C.

For mecillinam sensitivity titers, WT strains bearing either an empty vector or $^{Ec}$RodA-PBP2 variants on IPTG-inducible plasmids (MG1655/pRY47, MG1655/pHC857, MG1655/pSI154, MG1655/pSI157) were grown overnight in LB-CAM (permissive) at 37 °C to $OD_{600} = 0.4–0.8$ AU. The cultures were normalized for optical density, serially diluted, and spotted on LB-CAM plates supplemented with a range of sub-MIC concentrations of mecillinam (permissive, unless Rod complex activity is hyperactive) for overnight incubation at 37 °C.

### Phase contrast microscopy

For DTT rescue experiments, 3 mL overnight cultures of FB38/pHC857 and FB38/pSI47 strains in M9-malt-CAM supplemented with 25 μM IPTG were grown to $OD_{600} = 0.3–1$ AU at 30 °C. The resulting cultures were diluted 200- to 1000-fold into the same medium and grown to $OD_{600} = 0.1–0.2$; 100 μL of each culture was then spotted on a filter disk and allowed to incubate on LB-CAM-25 μM IPTG plates either with or without 10 mM DTT for 5 h at 30 °C. The resulting cultures were harvested, fixed with 2.6% formaldehyde/0.04% glutaraldehyde at room temperature for 1 h, and stored at 4 °C for up to 3 days prior to imaging. For imaging, cells were immobilized on 1.5% agarose pads containing M9 media and covered with #1.5 coverslips. Phase contrast microscopy was performed on a Nikon Ti2-E Inverted Motorized Microscope equipped with a Plan Apo 100x/1.45 Oil Ph3 objective and a 4.2bi Back Illuminated Cooled sCMOS camera (PCO). Images were acquired using Nikon Elements 5.2 Acquisition Software. Resulting images were processed to detect cells and quantify aspect ratios using MicrobeJ[55] (ImageJ2 version 2.9.0/1.53t), with all parameters set to default and segmentation enabled. At least 170 cells were analyzed for each condition, and statistical significance between conditions was assessed using a two-sample *t*-test (Matlab).

For $^{Ec}$MreC$^{R292H}$ rescue experiments, 3 mL overnight cultures of PR5/pHC857, PR5/pSI154, PR5/pSI157 were grown overnight in M9-glu-CAM at 30 °C. Overnight cultures were diluted 200- to 1000-fold into LB-CAM with varying concentrations of IPTG and grown to $OD_{600} = 0.3–0.6$ at 37 °C. The resulting cultures were harvested and

imaged as above. Image analysis was performed as described, following background subtraction using the rolling ball plugin (ImageJ) with radius set to 40 pixels to allow for accurate detection of spherical cells.

### Quantification of $^{Ec}$RodA-PBP2 expression levels in complementation strains

$^{Ec}$RodA-PBP2 expression levels in FB38/pHC857 and FB38/pSI47 strains were quantified using bocillin labeling as previously described[21]. Briefly, overnight cultures of FB38/pHC857 and FB38/pSI47 were grown in M9-malt media supplemented with kanamycin and chloramphenicol either with or without 250 μM IPTG to induce expression. Cultures were diluted 200-fold into 15 mL of the same media and grown to $OD_{600} = 0.4–0.6$. Equal amounts of total culture were collected for all conditions, and cell pellets were isolated by centrifugation (2000×$g$ for 10 min). After a wash with ice-cold 1xPBS (wash buffer), the pellets were respun and resuspended in 500 μL of 1xPBS-5mM EGTA and labeled with 20 μM bocillin for 15 min at RT. After three washes to remove excess bocillin, the cells were snap-frozen with liquid nitrogen and stored at −80 °C until ready for use. To isolate the membrane fraction, labeled cell pellets were resuspended in 500 μL of wash buffer supplemented with benzonase nuclease (Sigma Aldrich) at 1:100,000 dilution and EDTA-free protease inhibitors (Sigma Aldrich) and lysed using sonication. After a brief spin to remove undisrupted cells (4000×$g$ for 1–2 min), membrane fractions were pelleted by ultracentrifugation (200,000×$g$ for 30 min). The resulting membranes were resuspended in 100 μL solubilization buffer (1xPBS, 1% DDM, EDTA-free protease inhibitors), and total protein concentration was quantified using DC protein assay (Bio-Rad). For each condition, 20 μg of total protein was run on SDS-PAGE gel, and bocillin-labeled bands were visualized using a fluorescence imager (Amersham Typhoon 5).

### Protein expression and purification

**General procedure for expression and purification of Hisx6-SUMO-FLAG tagged proteins.** *Thermus thermophilus* RodA-PBP2 constructs were produced and purified as described previously[11]. Briefly, plasmids (pMS239, pSI7, pSI12, pSI13) were transformed into *E. coli* BL21 C43 cells, harboring a pAM174 (chloramphenicol, Ara) plasmid with SUMO-specific Ulp1 protease and grown on LB-agar plates supplemented with 50 μg/mL kanamycin (KAN) and 35 μg/mL chloramphenicol (CAM). Fresh transformants were used to inoculate 5 mL overnight cultures of LB-Kan-CAM. Overnight cultures were then diluted into 1 L of terrific broth (TB) media supplemented with the same antibiotics, as well as 2 mM MgCl$_2$, 0.1% glucose, and 0.4% glycerol, and grown to $OD_{600} = 2–3$ at 37 °C with vigorous shaking. Protein expression was induced with 1 mM IPTG and 0.1% arabinose at 18 °C. After 12–16 h of induction, the cultures were harvested by centrifugation and frozen with liquid nitrogen (LN) to be used directly or stored at −80 °C until future use.

Monomeric *E. coli* RodA-PBP2 constructs were produced by introducing mutations at the dimerization interface, swapping two hydrophobic residues for homologous hydrophilic residues in *T. thermophilus* (V143T-A147S). The resulting constructs (pSI126, pSI127, pSI147, pSI149, pSI152) were expressed essentially in the same way as *T. thermophilus* constructs, with one modification: 100 μg/mL ampicillin was swapped for kanamycin at all the relevant steps.

For protein purification, cell pellets were resuspended in 150 mL of lysis buffer per liter of culture (50 mM Hepes pH 7.5, 150 mM NaCl and 20 mM MgCl$_2$, benzonase nuclease at 1:100,000 dilution, EDTA-free protease inhibitors). After lysis by LM10 Microfluidizer (Microfluidics), cell fractions were isolated by centrifugation at 50,000×$g$ for 1 h and solubilized in 150 mL of resuspension buffer per liter of culture (50 mM Hepes pH 7.5, 350 mM NaCl, 10% glycerol, 1% n-dodecyl-β-D-maltoside). The samples were stirred for 1–2 h at 4 °C then centrifuged as before to isolate the soluble fraction. The resulting supernatant was supplemented with 2 mM CaCl$_2$ and loaded on anti-FLAG affinity resin

equilibrated with wash I buffer (resuspension buffer supplemented with 2 mM $CaCl_2$). The resin was subsequently washed with 5 column volumes (CV) of wash I buffer, followed by 10 CV of wash II buffer (20 mM Hepes pH 7.5, 350 mM NaCl, 5% glycerol, 0.1% DDM, 2 mM $CaCl_2$). Proteins were eluted with 0.2 mg/mL FLAG peptide and 5 mM EGTA and concentrated to 1–5 mg/mL using an Amicon concentrator with a 100 kDa molecular weight (MW) cutoff (Millipore Sigma). Protein samples were purified by size-exclusion chromatography (SEC) on the Superdex200 increase column (Cytiva) in a buffer containing 20 mM Hepes pH 7.5, 350 mM NaCl, and 0.1% DDM.

For polymerization assays, the same amount of total protein was loaded for all constructs to yield comparable elution profiles and concentrations after SEC (1 mg/mL), requiring little to no concentration for the assay. For cryo-EM, the main fractions of the monomeric peak were collected, concentrated, and subjected to an additional SEC purification step in 0.02% DDM. For smFRET imaging, eluted proteins were biotinylated and fluorescently labeled, and re-run on SEC in 0.1% DDM-containing buffer. All purified proteins were evaluated for purity by SDS-PAGE and for activity by the in vitro polymerization assay and then snap-frozen and stored at −80 °C until further use.

**Production and purification of the anti-ALFA nanobodies.** Anti-ALFA nanobody sequence[30,31] was cloned into the pET26b vector following an N-terminal periplasmic secretion signal (MKYLLPTAAAGLLLLAAQ-PAMA). ALFA variant compatible with Fab binding (pSI146) was produced by mutating two residues at the C-terminus (Q116K-Q119P) of the original anti-ALFA nanobody (pSI145) to match the TC-Nb4 scaffold sequence[29]. The introduction of these mutations did not affect the binding of the anti-ALFA nanobody to the ALFA tag.

Nanobodies were produced and purified as described previously[56]. Briefly, pSI145 and pSI146 plasmids were transformed into *E. coli* BL21 (DE3), plated on LB-Kan plates, and the resulting transformants were used to inoculate overnight 5 mL LB cultures. The overnight cultures were added to 250 mL of TB-Kan + 0.4% glycerol and allowed to grow at 37 °C to $OD_{600} = 1.5$–2, after which they were cooled gradually to 18 °C and induced overnight with 200 μM IPTG. Cells were pelleted at 4000×$g$ for 15 min and resuspended in 50–100 mL of room temperature SET buffer per liter of initial culture (200 mM Tris pH 8, 500 mM Sucrose, 500 μM EDTA). The cells were allowed to spin for 15 min and, once completely resuspended, diluted with 2x volume of cold $H_2O$, 5 mM $MgCl_2$, 1 μL benzonase, and stirred at room temperature for 1 h. The periplasmic supernatant was then isolated by centrifugation (9000–14,000×$g$ for 30 min) and supplemented with NaCl to 100 mM final concentration. After a 15-min stir at room temperature, the supernatant was loaded on 1–2 mL of Ni-NTA resin equilibrated with wash buffer I (1xHBS pH 7.5). The resin was washed with 10 CV of wash I buffer and 10 CV of wash II buffer (1xHBS pH 7.5, 10 mM imidazole), and protein was eluted with 10 CV of elution buffer (1xHBS pH 7.5, 200 mM imidazole). Eluted protein was concentrated to 5–10 mg/mL using an Amicon concentrator with a 3 kDa molecular weight (MW) cutoff and run on Superdex S75 (Cytiva) in 1xHBS pH 7.5, 2% glycerol buffer. Purity was checked by SDS-PAGE to ensure that the nanobody did not retain the PelB secretion sequence. The protein was snap-frozen and stored at −80 °C until further use in smFRET, polymerization, or binding assays.

**Production and purification of the nanobody-binding Fab.** The variable and human kappa constant regions of the NbFab light chain were cloned into the pD2610-v5 vector (ATUM) following an N-terminal (MDWTWRILFLVAAATGAHS) signal sequence. The variable and CH1 regions of the NbFab heavy chain were cloned into pTarget (Promega) following an N-terminal (MRPTWAWWLFLVLLLALWA-PARG) signal sequence.

A 250 mL culture of Expi293 cells at 3e6 cells/mL was transfected with 100 μg each of the heavy chain and light chain plasmids

(pMAS478, pMAS481) in 20 mL of Opti-MEM (Life Technologies) supplemented with 200 μL of FectoPRO (Polyplus). Following transfection, the culture was grown with shaking (125 RPM) at 37 °C with 8% $CO_2$. On the second day after transfection, the culture was supplemented with 300 mM valproic acid and 0.5% glucose and grown until day 7, when it was harvested. The supernatant containing the secreted Fab was isolated by centrifugation (4000×$g$ for 20 min) and applied to 1–2 mL of IgG-CH1 affinity resin (Thermo Fisher Scientific) pre-equilibrated with 1xHBS. The column was subsequently washed with 10 CV of 1xHBS. Protein was eluted with 7 mL of 100 mM citrate pH 3 directly into 3 mL of 1–2 M Hepes pH 8 per mL of resin to neutralize. The elution was dialyzed into SEC running buffer (1xHBS, 5% glycerol) buffer and purified on Superdex S75. Purity was checked by SDS-PAGE, and the protein was snap-frozen and stored at −80 °C until further use. The 1:1 nanobody:Fab complex was obtained by mixing the two proteins at 2–5 mg/mL and separating the complex from unbound components on the S75 column.

**Protein biotinylation and fluorescent labeling for smFRET experiments**
Purified proteins were labeled with 5–10 molar excess of sulfo-Cy3-maleimide and sulfo-Cy3-maleimide (Lumiprobe) in 20 mM Hepes pH 7.5, 350 mM NaCl, 0.1% DDM buffer for 10 min at room temperature. The reactions were quenched by the addition of 2 mM DTT and buffer-exchanged using Zeba spin desalting columns (Thermo Fisher Scientific) into biotinylation buffer (10 mM Tris-HCl pH 8, 50 mM NaCl, 0.1% DDM). The samples were biotinylated with 5 μg of BirA protein ligase (Avidity) per mg of protein for 1 h at room temperature. Following biotinylation, proteins were purified on S200I as described above, snap-frozen, and stored at −80 °C until further use.

**In vitro glycan polymerization assay**
Lipid II from *Enterococcus faecalis* was purified as described previously[57] and fluorescently labeled at the stem peptide lysine with Alexa Fluor 488 NHS ester (Life Technologies). For labeling, 100 μL of 500 μM lipid II in DMSO was mixed with 400 μL PBS pH 7.5 and 20 μL 0.2M sodium bicarbonate pH 9, and AF488 NHS-ester in DMSO was added in 10- to 20-fold molar excess of the lipid. The reaction was incubated for 1 h at room temperature and quenched with 50 μL of 1M Tris pH 8. Quenched reaction was then loaded on a C18 BakerBond column (Avantor) equilibrated with 2 mL each of 0.1% solution of ammonium hydroxide in methanol (MeOH/0.1% $NH_4OH$ v/v) and in water ($H_2O$/0.1% $NH_4OH$ v/v). The resin was washed with 1.5 mL of wash buffer, containing increasing concentrations of methanol (0, 20, 40, 60% MeOH/0.1% $NH_4OH$), and lipid II was eluted in two 750 μL fractions with 80–100% MeOH/0.1% $NH_4OH$. Elution fractions were dried, lyophilized, and resuspended in DMSO to a final concentration of 200 μM to be used in polymerization assays.

Polymerization reactions were adapted from protocols described previously[10,11]. Briefly, 10 μM stocks of *E. coli* or *T. thermophilus* RodA-PBP2 variants purified in 0.1% DDM buffer (see above) were diluted to 10 μL (1 μM final protein concentration) in 0.1% SEC buffer containing 20 μM lipid II-AF488 and 2 mM $MnCl_2$ and incubated for 30 min at 25 °C. The reaction was quenched by incubation at 4 °C and the addition of SDS loading dye. The samples were then loaded into a 4–20% gradient polyacrylamide gel and run at 200 V for 35 min, after which glycan chains were visualized using the fluorescence Typhoon imager (Amersham Typhoon 5).

All image analysis of polymerization gels was carried out in ImageJ (version 2.9.0/1.53t). The normalized total product for each condition was calculated by quantifying the total intensity of glycan chains, subtracting the background (lipid II-only control) and normalizing by the average intensity of the corresponding WT control. A two-sample $t$-test was used to determine whether each pair of constructs had statistically significant differences in the amount of total product,

assuming equal means but not necessarily equal variances in populations.

## Quantification of disulfide bond formation in $^{Ec}$RodA-PBP2$^{D49C-K240C}$

$^{Ec}$RodA-PBP2$^{D49C-K240C}$ (pSI47) was purified as described above, with all the buffers thoroughly degassed before use but no reducing agents added during purification. Following purification by size-exclusion chromatography, the fraction of disulfide-locked species was quantified using SDS-PAGE. Briefly, a fully reduced sample was run on an SDS-PAGE alongside an unreduced control. Given that disulfide crosslinked protein was found to migrate at a slower rate on SDS-PAGE than the reduced sample, the fraction of the disulfide-locked species could be determined by dividing the relative intensity of the top band (unreduced) by the intensity of the total protein (reduced). We note that as soon as the reducing agent was removed in vitro, the disulfide bond reformed spontaneously and with the same efficiency, indicating that in cells, the protein is disulfide-linked. The identities of the two bands were confirmed by liquid chromatography-mass spectrometry (LC-MS).

## Microfluidic chamber preparation and smFRET data collection

Microfluidic chambers containing coverslips functionalized with an 80:1 mixture of methoxypolyethylene glycol-succinimidyl valerate MW 5000 (mPEG-SVA-5000) and biotin-methoxypolyethylene glycol-succinimidyl valerate MW 5000 (biotin-PEG-SVA-5000 (Laysan Bio, Inc.) were prepared as described before and either used immediately or stored in a vacuum chamber until use for up to 7 days[58,59]. Solutions were applied to the chamber through a 1-mL syringe attached to PE60 tubing. Flow cells were first incubated with 20 μL of 1 mg/mL streptavidin in 1xPBS for 5–10 min (Sigma). Unbound streptavidin was flushed out with two 50 μL washes of SEC buffer (0.1% DDM, 300 mM NaCl, 20 mM Hepes pH 7.5 buffer), and fluorescently labeled and biotinylated protein samples were introduced into the chamber in the same buffer at 1–5 nM concentration. Unbound protein was washed out with 50 μL of SEC buffer and incubated prior to imaging with SEC supplemented with reactive oxygen-scavenging systems (ROXS): PCA (5 mM), PCD (0.1 μM), ascorbic acid (1 mM), methyl viologen (1 mM). We note that these additives had no effect on protein catalytic activity at the concentrations used in our assays.

Fluorescence images were acquired using a through-the-objective total internal reflection fluorescence (TIRF) microscope controlled by Hamamatsu HCImage live version 4.4.0.1 and Labview version 15.0f2, essentially as described before[58]. Unless specified otherwise, images were collected continuously for 150–300 s at a frame rate of 4 s$^{-1}$ (exposure time of 250 ms), alternating between two frames with excitation at 532 nm and one frame at 641 nm. Surface power densities were measured by a Coherent FieldMate power meter and set to 2 W/cm$^2$ (532 nm) and 1 W/cm$^2$ (641 nm). $^{Ec}$RodA-PBP2$^{L61R}$ and $^{Ec}$RodA-PBP2$^{T52A}$ were imaged at a faster frame rate (10 s$^{-1}$, corresponding to 100 ms exposure time) for 60–90 s to capture the faster dynamics of these mutants.

## Analysis of smFRET data

Analysis of single-molecule experiments was performed essentially as described previously using custom MATLAB scripts[58–61]. Fluorescence intensities were corrected for differences in Cy3/Cy5 quantum yield and detection efficiency for donor bleedthrough in the acceptor channel and direct acceptor excitation by the 532 nm laser as described[59,60]. Acceptor particles were picked by selecting intensity peaks in an averaged stack of the first five frames of the movie, then denoised and background subtracted. Corresponding donor particles were identified by iterative translation of the two channels in the *XY*-plane until maximum correlation was found. Donor and acceptor particles were matched if their centroids were displaced by no more

than 1.25 pixels, accounting for drift. 2D Gaussian fitting was performed on particle images from both channels, as well as the same images rotated 45°, and spots were accepted if their ellipticity scores (ratio of width to length) in both directions were 1–1.2 to exclude particles containing more than one fluorophore. Trajectories of filtered particles were truncated at photobleaching events of the donor or acceptor, and any trajectories containing more than one donor or acceptor were excluded from analysis (<10% of the total).

Resulting trajectories were further analyzed using the ebFRET GUI to quantify populations and identify FRET transitions in an unbiased manner (Supplementary Table 2)[62]. Dwell-time histograms for FRET states were generated by selecting time segments that were not censored by photobleaching from trajectories with at least one transition event. We note that dwell-time histograms were not reported for $^{Ec}$RodA-PBP2$^{L61R}$ and $^{Ec}$RodA-PBP2$^{T52A}$ mutants because these constructs had such fast dynamics that a large population of transition events were not resolved even at the higher frame rates (10 s$^{-1}$). We also excluded dwell-time fits for $^{Tt}$RodA-PBP2 since transitions in this sample are so rare that the apparent dwell times of the high-FRET state are limited by the photobleaching rate of the fluorophores. Transition frequency heat plots were generated by quantifying the total number of transition events (lasting more than a frame) observed in a dataset and normalizing by the total observation time. FRET-efficiency distribution plots, smFRET trajectory fits, single-exponential dwell-time histograms, and transition heat plots were generated using custom Matlab scripts.

The high- and low-FRET states were assigned to distinct structural conformations by comparing the inter-label distance estimates in the closed and open states of RodA-PBP2. In the crystal structure of the closed state (PDB: 6pl5), the distance is 30–40 Å, which corresponds to a high-FRET state (FE = 0.8–1), while in the AlphaFold model of the open state, the fluorophores are predicted to be 70–80 Å apart, consistent with low-FRET values (0.2–0.5). We note that the low-FRET population likely includes intermediate (partially open) states, along with the fully extended state, explaining why this population is broad and exhibits trajectories at 0.4–0.5 FE.

## Cryo-EM data collection and processing

The cryo-EM dataset for all samples was collected on a Titan Krios operating at 300 kV and equipped with a K3 camera. C-flat holey carbon 1.2/1.3 grids (Electron Microscopy Sciences) or UltrAuFoil holey gold film grids (Ted Pella) were plasma cleaned for 30 s using a Pelco Easiglow plasma discharge system set to 10 mA. Then, 4 μL of protein solution (Supplementary Table 2) prepared in 0.02% DDM, 350 mM NaCl and 20 mM HEPES pH 7.5 was deposited onto grids and plunge-frozen in liquid ethane using a Vitrobot Mark IV (Thermo Scientific) set to 100% humidity and 22 °C with a blot time of 7 s and a blot force of 15. Data were collected at ×105,000 magnification, corresponding to a calibrated pixel size of 0.825 Å. A total of 50–60 frames were collected for each micrograph for a total dose of ~60 e-/Å (Supplementary Table 2). Cryo-EM data processing, including motion correction, CTF estimation, blob-based picking, local motion correction, 2D classification, ab initio reconstruction, heterogeneous 3D refinement, and non-uniform 3D refinement, were performed in cryoSPARC v2 or v3[63–67].

For the $^{Tt}$RodA-PBP2$^{WT}$ dataset, 934,086 picked particles were subjected to local motion correction and iterative rounds of 2D classification, resulting in a particle stack of 249,246. Three rounds of 3D heterogeneous refinement resulted in a final particle stack of 93,553. Non-uniform refinement was used to generate the final 6.5 Å resolution map (Supplementary Fig. 3).

For the $^{Tt}$RodA-PBP2$^{A186R}$ dataset, 6,312,985 picked particles were subjected to local motion correction and iterative rounds of 2D classification, resulting in a particle stack of 247,426. Following ab initio reconstruction and heterogeneous refinement, two distinct 3D classes

were identified—open and closed—each containing 78,333 (open) and 92,393 (closed) particles. Non-uniform refinement was used to generate the final closed map at the corrected resolution of 6.1 Å. To obtain a map of the open state, open particles were subjected to non-uniform refinement (nominal resolution 4.5 Å), followed by 3D variability analysis, from which a single conformational snapshot was selected for the figure. Given that the fully open state constituted a fraction of the total conformational ensemble, estimates of the resolution were not reported for this state. Moreover, we note that the nominal resolutions of the open and closed ensembles from non-uniform refinement are likely overestimated due to overfitting (Supplementary Fig. 3).

For the $^{Ec}$RodA-PBP2$^{WT}$ dataset, 6,090,759 picked particles were subjected to local motion correction and iterative rounds of 2D classification, resulting in a particle stack of 141,314. Ab initio reconstruction and heterogenous refinement resulted in three distinct 3D classes: a minor monomeric class, a minor dimeric class (2:1 protein:micelle class), and a major dimeric class (1:1 protein:micelle). Only 1:1 dimer class had a sufficient number of particles (78,031) to yield a 3D map of sufficient quality (Supplementary Fig. 5). Non-uniform refinement of the 1:1 dimer resulted in a final map at 9 Å nominal resolution. Crystal structures of *E. coli* PBP2 either in the apo state (PDB: 6g9f) or in the MreC-bound conformation (PDB: 5lp5) were fit into the 3D map using the "fit in map" tool in Chimera (version 1.15), assuming a symmetric dimer[32,68,69].

### Reporting summary
Further information on research design is available in the Nature Portfolio Reporting Summary linked to this article.

## Data availability
Figure source data are provided with this paper. Uncropped SDS-PAGE gels and western blots from all the figures are provided in Supplementary Fig. 10. Supporting data are publicly available on Zenodo (https://doi.org/10.5281/zenodo.7884683). Source data are provided with this paper.

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

## Acknowledgements

We thank Marie Bao for critical reading of the manuscript and Clare Canavan and Daniel Kahne for generously sharing protocols for lipid II labeling. Cryo-EM data were collected at the Harvard Center for Cryo-Electron Microscopy and Harvard Medical School, and we thank them for their advice during data collection. We also thank the SBGrid Consortium for data processing resources and support. Phase contrast microscopy data were acquired in the Microscopy Resources on the North Quad (MicRoN) core at Harvard Medical School, and we thank them for providing support and resources during imaging experiments. We thank the Center for Macromolecular Interactions at Harvard Medical School for the use of instruments and support with data analysis. Funding for this work was provided by National Institutes of Health grants U19 AI158028 (to A.C.K. and T.G.B.), R01 AI153358 (to A.C.K.), R01 GM114065 (to J.J.L.), R01 AI148752 (to S.W.), R01 AI083365 (to T.G.B.), and Investigator funds from the Howard Hughes Medical Institute (to T.G.B.). I.S., E.M.F., and T.A.S. are supported by National Science (NSF) Foundation Graduate Research Fellowship awards. M.S.A.G. is

supported by a Hanna H. Gray Fellowship from the Howard Hughes Medical Institute.

## Author contributions

I.S., E.M.F., and M.S.A.G. performed and analyzed experiments. T.A.S. and S.W. contributed reagents. J.J.L., A.C.K., and I.S. conceived experiments and wrote the paper with input from M.S.A.G., E.M.F., and T.G.B.

## Competing interests

A.C.K. is a co-founder and consultant for Tectonic Therapeutic and Seismic Therapeutic and for the Institute for Protein Innovation, a non-profit research institute. The remaining authors declare no competing interests.
