## [Peer Review File · Nature Communications]

Allosteric activation of cell wall synthesis during bacterial growthREVIEWER COMMENTS

Reviewer #1 (Remarks to the Author):

The manuscript from Shlosman and colleagues is describing the molecular mechanism and dynamics of the PG synthase RodA-PBP2 from E. coli and T. thermophilus. The RodA-PBP2 synthase is an essential system for the biosynthesis of the peptidoglycan; the two enzymes catalyse the polymerisation and crosslinking of lipid II into glycan chains and polymerisation into existing peptidoglycan. Previous work from the authors had determined the structure of the RodA-PBP2 from T. thermophilus and key residues for the complex mechanism. In this study, they looked further into the structural basis of peptidoglycan formation and dynamics that govern the function of the synthase. They used smFRET to monitor the dynamics of the complex, which further informed their structural work by cryoEM. In brief, the smFRET shows that the complex exists mostly in a closed state with 'random' opening events. A combination of AlphaFold modelling and mutagenesis at the hinge region of the complex resulted in an open structure that was further validated by cryoEM and in vivo bacterial cell growth assays. They provide some evidence that the activity of the RodA-PBP2 complex is regulated by the MreC protein that can allosterically open the synthase as it is predominantly closed (as shown by smFRET).

Overall this is an excellent study with careful controls and novel insights in the function of the RodA-PBP2 complex. This work is worth of publication in Nature communications.

I only have a few suggestions/comments for the authors:

The smFRET experiments have measured the dynamics of the complex in the absence of a substrate. Did the authors attempt to include substrates in their studies that could have provided more insights on the progress of the polymerisation including release of the product?

Regarding the smFRET experiments, it is unclear how the FRET efficiency was 'mapped' onto the structures as a possible conformation; ie did they use any molecular rulers of known distances or did they assume/estimate the FRET efficiency by looking at the available structures? It needs better clarification in the methods and main body.

Regarding the E. coli RodA-PBP2 dimer, what's the evidence that it does not exist in the E. coli membrane?

In respect to the MreC experiments, did the authors try to make the truncated version of the protein as in Xu et al FEBS (<https://doi.org/10.1002/2211-5463.13296>) that could provide further evidence on its role in regulating the complex?

Reviewer #2 (Remarks to the Author):

Re-Irina Shlosman and company address a fundamental question in the field of bacteriology and that is how protein complexes synthesize the peptidoglycan (PG). In this manuscript the authors use single-molecule FRET and cryo-EM to show that an essential PG synthase (RodA-PBP2), an important step in bacterial elongation, undergoes dynamic exchange between closed and open states. To date, although many complexes involved in PG synthesis have been identified there is very little known about how these proteins come together to synthesize the PG. The authors also complement their finding with biochemical and in vivo data showing that the opening of the RodA-PBP2 complex is also coupled to the polymerization and crosslinking of the PG. The authors suggest the opening motion of the RodA-PBP2 complex likely represents a conserved regulatory mechanism that controls activation of PG synthesis during cellular processes, such as cell division.

In my view the authors covered extensive and comprehensive ground answering how this PG polymerase works. I really enjoyed reading this very long paper because as soon as I had a question it was answered shortly thereafter in the upcoming paragraphs. I have no major critics that would result in additional experiments. My comments below will be minor and to improve the clarity of the paper for people outside the field.

a) Mrec appears to control the activity of the PG polymerase and therefore is a key player in the overall mechanism. I would add a few lines about MreC in the introduction specifically about what is known in the field in reference to PBP2.

b) 'More generally, the finding that single amino-acid substitutions at interface II dramatically alter the conformational ensemble of a SEDS- bPBP—and by extension, the enzyme's activity—suggests that this allosteric switch represents a powerful evolutionary adaptor that can be tuned or repurposed to satisfy diverse molecular needs', This section needs further clarity in the way it is written. Especially the part on evolutionary adaptors that could satisfy diverse molecular needs.

c) Figure 6 along with the legend is quite dense. Anyway, it could be modified for clarity and readability.

d) Do the authors suggest that a monodisperse preparation of EcMreC is necessary for interaction with the entire RodA-PBP2 complex. Since EcMreC is also known to form multimeric EcMreC and would have to transition to a monodisperse EcMreC to facilitate the function of RodA-PBP2 complex. How does the authors think this occurs?

e) The authors conclude that the open state of EcRodA-PBP2 is required for PG synthesis in vivo. In the cryo-EM experiments of wild type TtRodA-PBP2 or other mutants, did the authors notice potential intermediate positions between the open and closed position during 2D or 3D classification even as

minor species? It was not clear to this reviewer what a partial lock could accomplish?... If the partial lock was artificial, or is it a necessary, part of the transition state that must go from open to close that could facilitate a pause such that the complexes may move along efficiently to polymerize the PG strand where other protein players are also at work eg. the hydrolases.

f) Going through the relative constructs related to EcRodA-PBP2 and TtRodA-PBP2 WT was sometimes confusing.

Reviewer #3 (Remarks to the Author):

Shlosman et al. present a very sound study on the regulation of the last step in the biosynthesis of peptidoglycan (PG), the main component of the bacterial cell wall. There are two primary mechanisms; one of them is performed by the bi-functional class A PBPs and the other is performed by the membrane complex between the SEDS proteins (like RodA), responsible for polymerization of the glycan chains, and the mono-functional class B PBPs responsible for transpeptidation of the peptide stems attached to the glycan chains. In previous works by the same team, the 3D structure of the RodA-PBP2 was reported providing the first indication of how the SEDS proteins recognize PBPs and how the glycan and peptide polymerization could be coordinated. A surprising result of that previous work was that PBP remained close to the membrane with the active site far from the PG; thus, pointing to dynamic behavior. This is the main goal of the present manuscript.

By a clever combination of FRET and cryoEM techniques, the authors conclude the presence of two states, closed (Inactive) and open (active), in such a way that glycan polymerization and transpeptidation are performed synchronously in the open state; and contrary to previously hypothesized by the same authors (in which closed state would perform polymerization and open state transpeptidation). Considering the relevance of this mechanism, the present work presents very valuable information that, with some adaptations, could be applied to most bacterial species.

While the manuscript is very well written, there are some potential improvements to clarify critical aspects of the work, as well as some errors to be corrected as detailed below.

Main Points:

1) There is a complete mismatch between the Supplemental figures/tables and the names in the main text which makes it very difficult for reading (I was expending much time trying to understand the relationship between supplemental figures and the main text). Please correct it.

2) While the cryoEM provides valuable information about the two different states for the RodA-PBP2 system, unfortunately, no high/medium resolution was provided for the open state due to conformational dynamics. However, based on AF predictions authors identify a triad of residues in

ecPBP2 that keep the PBP in a fully active conformation. Thus, is there no way to try these mutations in the whole RodA-PBP2 complex to get a high-resolution cryoEM model of the complex? this information would be extremely important to understand in detail the proposed allosteric mechanism as well as to identify the potential connection between the GT and TP processes.

3) When providing a model for the regulation of RodA-PBP2 by MreC it is worth calculating the putative ternary complex by using AF. This could provide extra information very useful to understand how this process could take place.

Minor points:

1) In the introduction section when describing the differential role of class A PBPS, include also the following reference:

Daniel Straume et al. Class A PBPs have a distinct and unique role in the construction of the pneumococcal cell wall. *Proceedings of the National Academy of Sciences, PNAS* (2020) 117, 6129-6138 (doi:/10.1073/pnas.1917820117)

2) In the "Hinge and interface mutations promote structural dynamics in TtPBP2" section. When calling Fig. 1A to describe the "tight H-bond network", this figure does not give any information about such an H-bond network. Please provide it. By the way, from the deposited structure (PDB code 6PL5) I see that only R197 is involved in H-bonds but not R37. I don't know if authors manage other information justifying mutation at that site (maybe just its position at the hinge); please explain.

3) When using AF to explore change at the hinge region between closed and open states. Please add a supplemental figure to see the change in the H-bond network between both states. The hinge is pivotal to understanding the change from the closed to the open conformations.

REVIEWER COMMENTS

We thank the reviewers for their thoughtful assessment of the manuscript and for the constructive feedback that helped improve this study. We have revised our manuscript as recommended and provided responses to the reviewers' questions and comments below. Changes to the text are indicated in the manuscript for clarity.

Reviewer #1 (Remarks to the Author):

The smFRET experiments have measured the dynamics of the complex in the absence of a substrate. Did the authors attempt to include substrates in their studies that could have provided more insights on the progress of the polymerisation including release of the product?

We agree with the reviewer that this experiment would be very interesting, as it would provide additional insight into the role of structural dynamics in lipid II polymerization. We have attempted experiments where lipid II (with or without divalent cations) was added to the reaction and smFRET dynamics were measured. However, we didn't observe a significant change in the smFRET distributions, likely because lipid II did not bind the protein efficiently in the context of the flow cell. Using fluorescently labeled lipid II, we confirmed that lipid II preferentially binds the surface of the flow cell, instead of co-localizing with the protein. Thus, without considerable further optimization of the experimental set-up, our smFRET assay cannot assess the effect of lipid II on dynamics.

Regarding the smFRET experiments, it is unclear how the FRET efficiency was 'mapped' onto the structures as a possible conformation; ie did they use any molecular rulers of known distances or did they assume/estimate the FRET efficiency by looking at the available structures? It needs better clarification in the methods and main body.

To assign FRET states to distinct conformations, we relied on structures and structure predictions, rather than using a molecular ruler. Our cryo-EM analysis of the WT protein and the dynamic mutant (Figure 2) identified two conformations: one extended and one closed. The closed state largely resembles previously resolved X-ray structure, in which the labels are positioned within ~ 30 Å of each other. From this, we inferred that the closed conformation must correspond to the high-FRET state (0.8-1). The resolution of the extended conformation is not high enough to get a precise estimate of the distance between the labels. However, since this state is similar to the AlphaFold prediction of the protein, we used the AlphaFold model to estimate the inter-label distance in the extended state (~ 70 Å). This distance is consistent with the broad low-FRET population (0.2-0.5), which, presumably, corresponds to the fully extended state as well as any intermediate conformations. Since there were only two smFRET states and two structural states, we were confident that our assignment was correct, without further validation with a molecular ruler.

We have now added a clarification in the figures (Figure 1), the main text (line 98; line 144) and the methods (lines 751-757) that explains how conformational assignment was done.

Regarding the Ecoli RodA-PBP2 dimer, what's the evidence that it does not exist in the Ecoli membrane?

While it is possible in principle that *E. coli* RodA-PBP2 dimerizes in a physiological context, the dimerization that we observed in our sample is most likely non-native. Supplementary Figure 4B shows that protomers in the major dimeric population interact through the pedestal domain of

PBP2, so as to position the TM domains at a 90° angle with respect to each other. This geometry would not be compatible with the dimer being embedded in a continuous lipid bilayer.

In respect to the MreC experiments, did the authors try to make the truncated version of the protein as in Xu et al FEBS (<https://doi.org/10.1002/2211-5463.13296>) that could provide further evidence on its role in regulating the complex?

We thank the reviewer for reminding us of this study. We have not tested this truncated construct yet, but will explore this as a candidate in future smFRET experiments.

Reviewer #2 (Remarks to the Author):

a) Mrec appears to control the activity of the PG polymerase and therefore is a key player in the overall mechanism. I would add a few lines about MreC in the introduction specifically about what is known in the field in reference to PBP2.

We have added more background about MreC in the introduction (lines 49-52; lines 59-61)

b) 'More generally, the finding that single amino-acid substitutions at interface II dramatically alter the conformational ensemble of a SEDS- bPBP—and by extension, the enzyme's activity—suggests that this allosteric switch represents a powerful evolutionary adaptor that can be tuned or repurposed to satisfy diverse molecular needs', This section needs further clarity in the way it is written. Especially the part on evolutionary adaptors that could satisfy diverse molecular needs.

The diverse molecular needs that we are referring to are distinct modes in which PG synthesis is activated and regulated in different bacteria (e.g., reliance on MreC/D for activation vs use of alternative accessory components; higher or lower levels of basal activity of the RodA-PBP2 complex etc). We speculate that bacteria can alter or repurpose their regulatory mode with only a few mutations by altering the structural dynamics of PBP2 or coupling those dynamics to activation by a different adaptor protein. To make this point clearer, we have reordered the discussion (lines 409-415).

c) Figure 6 along with the legend is quite dense. Anyway, it could be modified for clarity and readability.

In view of this comment, we have divided Figure 6 into two figures, making the mechanistic model into a separate figure to improve clarity and readability (Fig. 7).

d) Do the authors suggest that a monodisperse preparation of EcMreC is necessary for interaction with the entire RodA-PBP2 complex. Since EcMreC is also known to form multimeric EcMreC and would have to transition to a monodisperse EcMreC to facilitate the function of RodA-PBP2 complex. How does the authors think this occurs?

We thank the reviewer for letting us clarify this point. For the purposes of smFRET experiments, we attempted to produce a well-behaved construct of MreC (dimeric or monomeric) that would bind RodA-PBP2 stably, since MreC likely exists as either monomer or dimer in the Rod complex. However, as the reviewer points out, MreC has been previously proposed to form clusters in cells (Martins *et al.* Nature Comm 2021). In this study, the authors speculate that changes in the oligomeric state of MreC play a regulatory role: MreC dissociates into individual monomers/dimers upon incorporation into the Rod complex, through its contacts with PBP2. It's

an interesting mechanism, and would potentially implicate PBP2 dynamics in regulating the oligomeric state of MreC and in its recruitment to the complex.

e) The authors conclude that the open state of EcRodA-PBP2 is required for PG synthesis *in vivo*. In the cryo-EM experiments of wild type TtRodA-PBP2 or other mutants, did the authors notice potential intermediate positions between the open and closed position during 2D or 3D classification even as minor species? It was not clear to this reviewer what a partial lock could accomplish?... If the partial lock was artificial, or is it a necessary, part of the transition state that must go from open to close that could facilitate a pause such that the complexes may move along efficiently to polymerize the PG strand where other protein players are also at work eg. the hydrolases.

We could not distinguish intermediate positions between open and closed states by cryo-EM with sufficient confidence due to the high conformational heterogeneity of the sample. For that reason, we resolved only the fully open (stable) state of the conformational ensemble. Our smFRET experiments suggest the presence of conformational intermediates: binding of the Nb-Fab (Fig. 4), or mutations in the hinge region (Fig. 6) induce conformational transitions into 2 states (low-FRET, or fully open, and middle-FRET, or partially open), rather than a single state.

It is certainly likely that these partially open states represent intermediates on the path to opening, while the fully open state is the activated state. We expect that upon interaction with the PG substrate *in vivo*, all conformers transition into the fully extended conformation. Thus, the effect of mutations that induce partial opening is to lower the energy barrier to the opening transition, increasing the frequency of enzymatic activation. That said, we agree with the reviewer that intermediate states may also play distinct roles in catalysis. This would be a very interesting question to resolve; unfortunately, we do not currently have a structure-function assay that would allow us to relate individual steps of the polymerization reaction to distinct conformational states.

Finally, we'd like to clarify the meaning of the "partial lock" that the reviewer is alluding to, in reference to the *in vivo* experiments in Fig. 5. In those experiments, we "lock" the protein in the closed state with disulfide crosslinks. This locked population cannot transition to the open state (or any half-open states), leading to enzymatic silencing. However, since only 50% of the protein is disulfide-locked (the rest is free to undergo dynamics), we call this strategy a "partial lock". We believe that the residual Rod complex activity that we observe in the strain with the "partially locked" mutant is due to the fact that 50% of the protein is still active.

To improve clarity, we reworded this part of the text to omit the expression "partial lock".

f) Going through the relative constructs related to EcRodA-PBP2 and TtRodA-PBP2 WT was sometimes confusing.

We have added a Supplementary Figure (Supplementary Fig. 2), showing the sequence alignment between EcPBP2 and TtPBP2 to clarify the mapping of different mutations.

Reviewer #3 (Remarks to the Author):

Main Points:

1) There is a complete mismatch between the Supplemental figures/tables and the names in the

main text which makes it very difficult for reading (I was expending much time trying to understand the relationship between supplemental figures and the main text). Please correct it.

We thank the reviewer for catching this mistake. We have corrected the numbering in the revised manuscript.

2) While the cryoEM provides valuable information about the two different states for the RodA-PBP2 system, unfortunately, no high/medium resolution was provided for the open state due to conformational dynamics. However, based on AF predictions authors identify a triad of residues in ecPBP2 that keep the PBP in a fully active conformation. Thus, is there no way to try these mutations in the whole RodA-PBP2 complex to get a high-resolution cryoEM model of the complex? this information would be extremely important to understand in detail the proposed allosteric mechanism as well as to identify the potential connection between the GT and TP processes.

We agree that having a high-resolution structure of the open state of the RodA-PBP2 complex would provide invaluable insight into the structural rearrangements that underlie enzymatic allostery. We attempted to resolve the structure of the triple *E. coli* mutant using cryo-EM. Unfortunately, the *E. coli* protein wasn't sufficiently well-behaved and exhibited aggregation upon freezing, which precluded its direct use in cryo-EM. Extensive sample optimization falls outside of the scope of this study, but we hope to improve the behavior of this mutant and determine its structure in the future.

3) When providing a model for the regulation of RodA-PBP2 by MreC it is worth calculating the putative ternary complex by using AF. This could provide extra information very useful to understand how this process could take place.

We have added a Supplementary Figure (Supplementary Fig. 7), showing AlphaFold-predicted interactions within the ternary complex between *E. coli* RodA, PBP2 and MreC (monomeric).

Minor points:

1) In the introduction section when describing the differential role of class A PBPS, include also the following reference:

Daniel Straume et al. Class A PBPs have a distinct and unique role in the construction of the pneumococcal cell wall. Proceedings of the National Academy of Sciences, PNAS (2020) 117, 6129-6138 (doi:/10.1073/pnas.1917820117)

We thank the reviewer for reminding us of this study. We have added the reference to the introduction.

2) In the "Hinge and interface mutations promote structural dynamics in TtPBP2" section. When calling Fig. 1A to describe the "tight H-bond network", this figure does not give any information about such an H-bond network. Please provide it. By the way, from the deposited structure (PDB code 6PL5) I see that only R197 is involved in H-bonds but not R37. I don't know if authors manage other information justifying mutation at that site (maybe just its position at the hinge); please explain.

We have altered Fig. 1A to include a depiction of the hydrogen-bond network. We thank the reviewer for pointing out a mistake in the numbering: the hinge mutant is in fact R33A-R197A

(not R37A-R197A), with both of the arginines engaged in hydrogen bonding in the hinge region. We have corrected this error throughout the manuscript.

3) When using AF to explore change at the hinge region between closed and open states. Please add a supplemental figure to see the change in the H-bond network between both states. The hinge is pivotal to understanding the change from the closed to the open conformations.

We added a supplementary figure (Supplementary Fig. 3) that shows the loss of key hinge contacts in the AlphaFold model of the open state.